# A pan-cancer genome-wide analysis reveals tumour dependencies by induction of nonsense-mediated decay

Zhiyuan Hu[1,2,3], Christopher Yau[3,4] & Ahmed Ashour Ahmed[1,2]

Nonsense-mediated decay (NMD) eliminates transcripts with premature termination codons. Although NMD-induced loss-of-function has been shown to contribute to the genesis of particular cancers, its global functional consequence in tumours has not been characterized. Here we develop an algorithm to predict NMD and apply it on somatic mutations reported in The Cancer Genome Atlas. We identify more than 73 K mutations that are predicted to elicit NMD (NMD-elicit). NMD-elicit mutations in tumour suppressor genes (TSGs) are associated with significant reduction in gene expression. We discover cancer-specific NMD-elicit signatures in TSGs and cancer-associated genes. Our analysis reveals a previously unrecognized dependence of hypermutated tumours on hypofunction of genes that are involved in chromatin remodelling and translation. Half of hypermutated stomach adenocarcinomas are associated with NMD-elicit mutations of the translation initiators *LARP4B* and *EIF5B*. Our results unravel strong therapeutic opportunities by targeting tumour dependencies on NMD-elicit mutations.

[1] Weatherall Institute of Molecular Medicine, University of Oxford, Oxford OX3 9DU, UK. [2] Nuffield Department of Obstetrics and Gynaecology, University of Oxford, Oxford OX3 9DU, UK. [3] Wellcome Trust Centre for Human Genetics, University of Oxford, Oxford OX3 7BN, UK. [4] Centre for Computational Biology, Institute of Cancer and Genomic Sciences, University of Birmingham, Birmingham B15 2TT, UK. Correspondence and requests for materials should be addressed to C.Y. (email: c.yau@bham.ac.uk) or to A.A.A. (email: ahmed.ahmed@obs-gyn.ox.ac.uk).

Nonsense-mediated decay (NMD) is a physiological cellular surveillance system that degrades abnormal mRNAs with premature termination codons (PTCs)[1]. PTCs are known to be involved in about one-third of all genetic diseases[2] and in cancer. Because NMD leads to significant loss of expression[3], it results in loss-of-function (LoF) of the affected genes and, therefore, a considerable contribution to the genesis of diseases[4]. An example of this is how the inactivation of tumour suppressor genes (TSGs) harbouring PTCs through NMD is thought to contribute to cancer initiation[5–7]. Therefore, the inhibition of NMD has been proposed as a promising therapeutic modality in cancer[4]. However, the extent to which NMD contributes to cancer progression has remained poorly understood as this requires the systematic identification of genes with PTCs and knowledge of the rules that govern whether or not NMD is likely to occur. Studies have focused predominantly on whether single-nucleotide variants were predicted to generate a PTC but have not attempted to predict the effect of frameshift mutations. For example, a recent study showed that TSGs harboured more PTCs that resulted from single-nucleotide variants leading to stop codons compared to other genes[8]. Recent work has validated the previously known rules for predicting whether a mutation is likely to elicit NMD (NMD-elicit) or not (NMD-escape)[5], but the global impact of NMD-elicit mutations on cancer has remained unexplored.

In this work we used three recently validated rules[5] to predict whether a mutation that may result in a PTC is NMD-elicit or NMD-escape (Fig. 1a). For a mutation to elicit NMD, three conditions are thought to be required. The first condition is that the PTC should be more than 50–54 bp upstream of the last exon–exon junction[9]. The second is that the targeted gene has to comprise at least two exons[10], and the last is that the PTC should be more than 200 bp downstream of the start codon. These three rules can explain up to 80% of the decreased expression of the mutated genes[5]. The analysis of over 1 million somatic mutations across 24 cancers from The Cancer Genome Atlas (TCGA) predicts 73,855 NMD-elicit mutations and provides a comprehensive landscape of NMD targeting in 7,725 genomes and corresponding transcriptomes. NMD compromises the expression of mutated TSGs, which may facilitate the initiation or progression of cancers. In contrast, NMD-elicit mutations in non-TSGs cluster in particular pathways to promote a phenotypic effect.

## Results

### Pan-cancer analysis for discovery of NMD-elicit mutations.
To classify TCGA mutations to NMD-elicit, NMD-escape or others, we developed a prediction algorithm based on the three afore-mentioned rules of eliciting NMD (Fig. 1a,b). For each mutation, the algorithm defined the open reading frames (ORFs) of the affected gene, located the first termination codon in the ORF and computed the distance of the codon from the last exon–exon junction. We applied our prediction algorithm to all reported somatic mutations from 24 cancers and predicted 73,855 (6%) NMD-elicit mutations (Supplementary Data 1). Unexpectedly, our analysis indicated that only two-thirds of the mutations that were annotated by TCGA as nonsense mutations (NMs) or frameshift indels were actually predicted to be NMD-elicit mutations (Fig. 1c).

To validate our classifier, we next compared the expression levels of NMD-elicit frameshift or NMs with other mutations. Overall, the predicted NMD-elicit mutations had significantly lower expression compared to NMD-escape, non-PTC-harbouring or silent mutations (median ratio of relative expression of variant (REV; see Methods) = 0.54, $P < 2.2e - 16$, one-sided $t$-test and Fig. 1d). Importantly, NMD-escape mutations were not associated with a reduction in the expression of the affected genes compared to silent mutations (median ratio of REV = 1, $P = 0.1$, one-side

$t$-test). Of note, stomach adenocarcinoma (STAD), kidney cancer (KIRP) and colon cancer (COAD) had a disproportionately higher number of NMD-elicit mutations, compared to other cancers with similar mutation frequency ($P < 0.007$, by generalized linear regression; Fig. 1e). These results strongly suggest that NMD plays an important role in contributing to the LoF of the affected genes in cancer by loss of expression.

### TSGs are frequently inactivated by NMD.
We observed that not all NMD-elicit mutations were associated with a similar magnitude of reduction in gene expression. In addition, NMD-elicit mutations seemed to target particular genes at higher frequency than others. For example, the two genes that were most widely affected by NMD-elicit mutations were the TSGs *TP53* (23 cancer types affected) and *NF1* (22 cancer types affected). However, the frequency of NMD-elicit mutations appears to be lower than other LoF mutations. For example, NMD-elicit *TP53* mutations occurred in 8.8% of all cancer samples ($n = 7,725$), while missense non-synonymous mutations in the same gene occurred in 22.2% of samples. NMD-elicit mutations in these genes are associated with a significant reduction in gene expression (median ratio of REV = 0.07 for *NF1*, 0.06 for *TP53*; $P < 2.2e - 16$, one-side Mann–Whitney–Wilcoxon (MWW) test and Fig. 2a). To formally evaluate these two factors (frequency and the magnitude of NMD-associated reduction in expression), we developed a score to measure the magnitude of NMD-associated reduction in expression ($z$-score, see Methods). We conducted a global analysis of all reported mutations in TCGA and observed that the NMD-elicit mutations that had the lowest $z$-scores (highest magnitude of reduction) and the most frequency tended to occur in known TSGs or the previously reported significantly mutated genes (SMGs) in cancer[11] (Supplementary Fig. 1). The overall prevalence of NMD-elicit mutations in TSGs[12] was 29% (2,206/7,725; range: 5–60%). In uterine corpus endometrioid carcinoma (UCEC), bladder urothelial carcinoma (BLCA) and stomach cancer (STAD), more than half of cases had NMD-elicit mutations within TSGs (Fig. 2b). The list of TSGs or SMGs with highly frequent NMD-elicit mutations varied according to the tumour types (Fig. 2c and Table 1). For example, the *ATRX* gene frequently harbours NMD-elicit mutations in low grade glioma and sarcoma. *APC* is often affected by NMD-elicit mutations in colon cancer (COAD) and rectal cancer (READ). Of particular note is that BLCA is uniquely affected by *KDM6A* NMD-elicit mutations when compared with other tumour types.

Overall, NMD-elicit mutations in TSGs were associated with significantly lower $z$-scores compared to NMD-elicit mutations in other genes (mean ratio = 0.29/0.39, $P < 2.2e - 16$, one-sided $t$-test, Supplementary Fig. 2). To evaluate the potential reason for this, we tested the co-occurrence between NMD-elicit mutations and chromosomal deletions at the affected loci. We observed a strong association between NMD-elicit mutations in TSGs and deletions in the same locus (Supplementary Fig. 3a). Thus, the profound reduction in gene expression at TSGs is most probably the result of NMD of the mutated allele and deletion of the wild-type allele. Importantly, in the absence of a deletion of an allele, NMD-elicit mutations are still associated with a significant reduction of gene expression but at a lower magnitude compared to cases when a deletion is present ($P < 2.2e - 16$, MWW test, fold change = 0.43 and $P < 2.2e - 16$, MWW test, fold change = 0.09, respectively, Supplementary Fig. 3b).

### NMD-elicit mutations associated with hypermutation.
Our analysis, thus far, suggests that NMD-elicit mutations at TSGs are associated with profound loss of expression because of associated deletions of the wild-type allele. In contrast, NMD-elicit

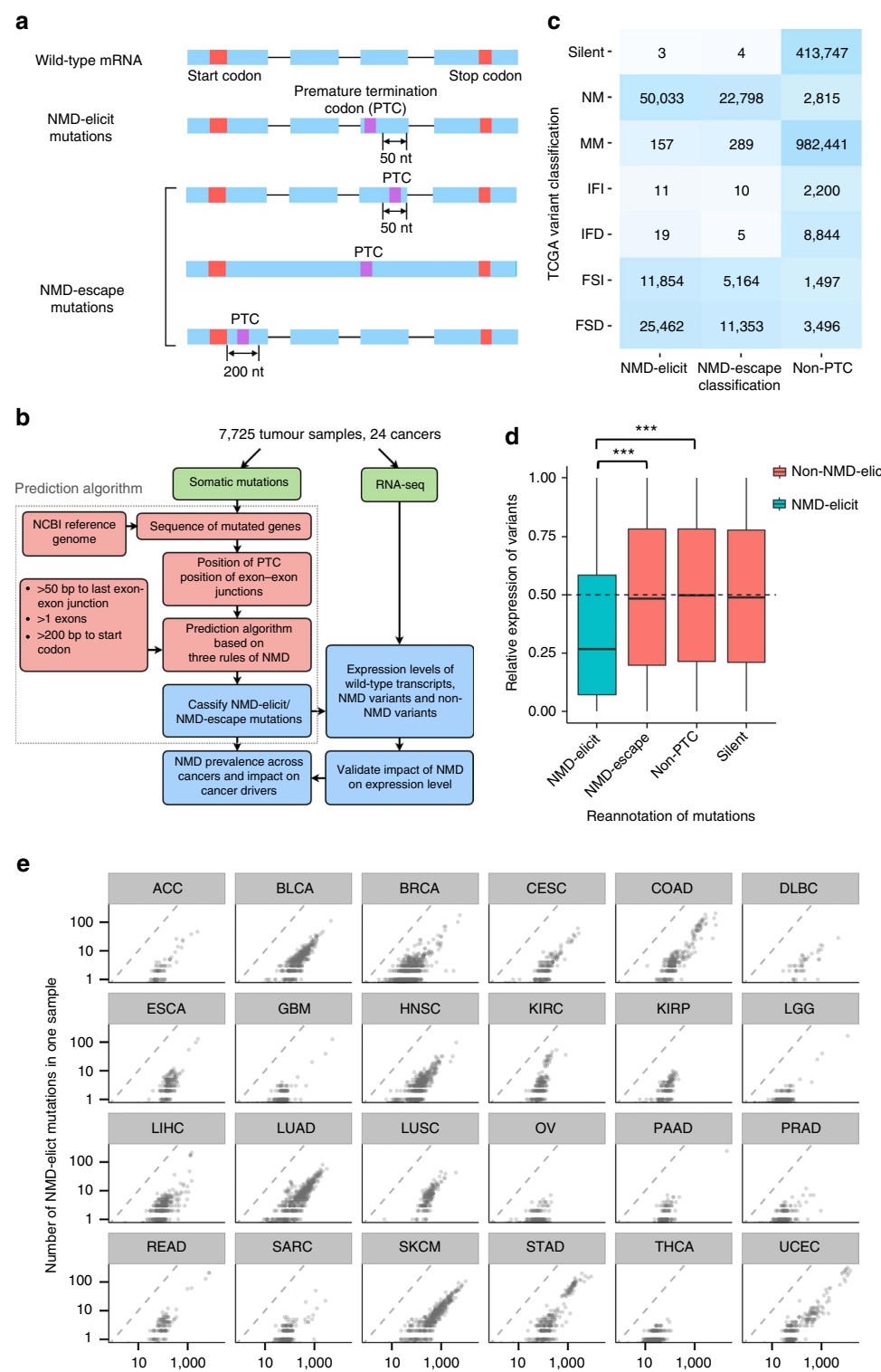

**Figure 1 | The landscape of NMD-elicit mutations in cancer.** (**a**) A diagram showing the three rules used to annotate NMD-elicit mutations. (**b**) A schematic showing the pipeline used for the prediction of NMD-elicit mutations in TCGA data. (**c**) A summary of the number of NMD-elicit mutations categorized by their original TCGA classification as indicated. FSD, frameshift deletion; FSI, frameshift insertion; MM, missense mutation; NM, nonsense mutation; Silent, silent mutation; IFD, in-frame insertion; IFI, in-frame insertion. (**d**) Boxplots comparing the expression levels of genes that harbour NMD-elicit (green) and NMD-escape (red) frameshift indels and nonsense mutations. The horizontal line at 0.5 indicates no differential expression. The expressions in the NMD-elicit group ($N = 53,406$) are significantly lower than those in the NMD-escape ($N = 19,773$), non-PTC-harbouring ($N = 488,430$) or silent ($N = 202,508$) groups (median ratio of REV = 0.54; \*\*\*$P < 2.2e - 16$, one-sided $t$-test). (**e**) Scatter plots showing the correlation between the number of all mutations in a sample and the number of NMD-elicit mutations for each tumour type as indicated. The grey dashed lines are the angle bisectors of the first quadrants.

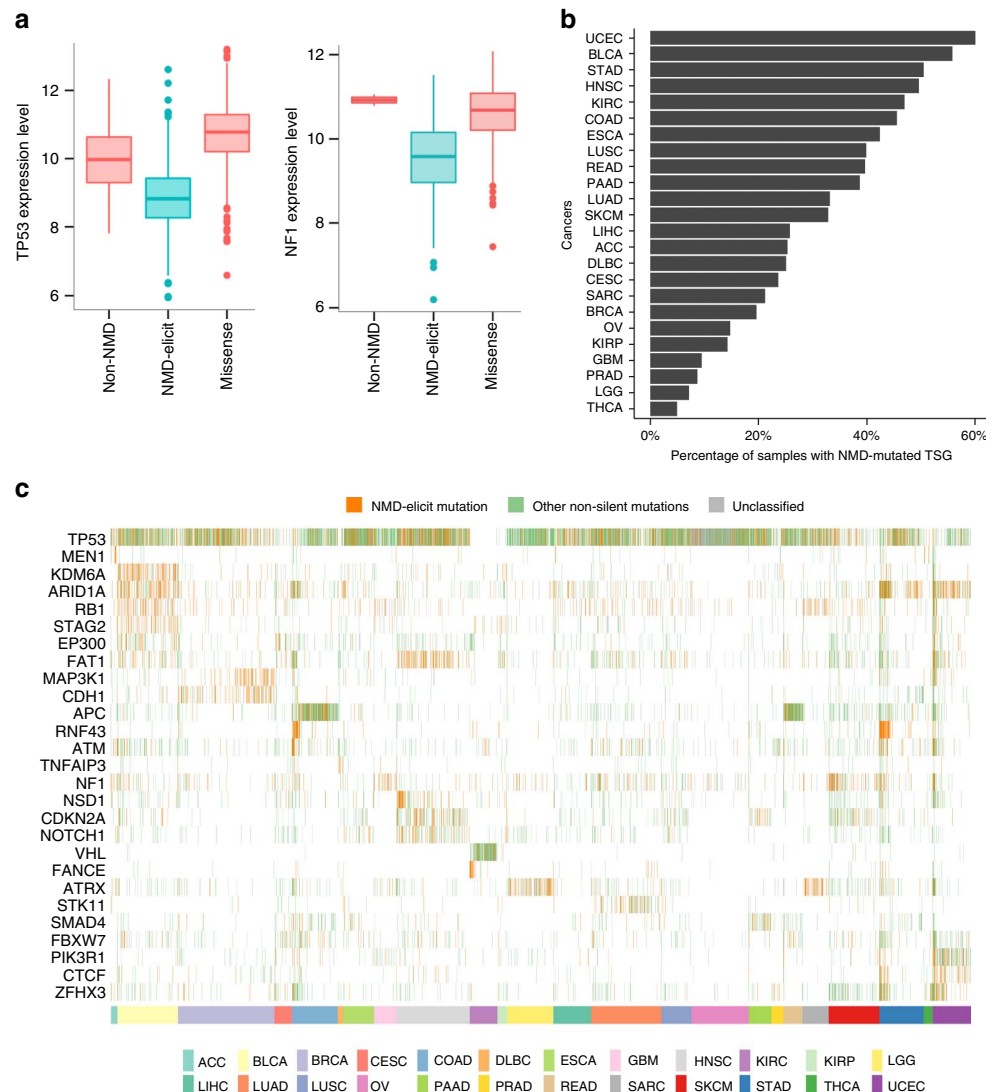

**Figure 2 | Tumour type-specific signatures for NMD-elicit mutations in tumour suppressor genes.** (**a**) Boxplots showing the expression levels of the indicated genes categorized by the type of mutations. NMD-elicit includes NMD-elicit frameshift insertions/deletions (indels) and nonsense mutations; non-NMD includes frameshift and nonsense mutations that are not NMD-elicit. The genes harbouring NMD-elicit mutations have significantly lower expression than the genes harbouring non-NMD or missense mutations (median ratio of REV = 0.07 for *NF1*, 0.06 for *TP53*; $P < 2.2e - 16$, one-side MWW test). The missense is the missense mutation. For *TP53*, the numbers in each group correspond to 39, 642 and 1,675. For *NF1*, the numbers in each group correspond to 2, 171 and 177. (**b**) A barplot representation of the percentage of samples harbouring NMD-elicit TSGs (x axis) across cancer types (y axis). (**c**) A heatmap indicating mutations of genes (rows) across samples (columns) as NMD-elicit (orange), non-NMD-elicit and non-silent (green) or others (grey). Samples were organized by cancer type as indicated in the legend.

mutations in non-TSG genes were associated with a modest loss of expression and occurred at low frequency and are, therefore, most probably associated with hypofunction of the affected gene. While single mutations of that type are unlikely to have important functional consequences, several mutations targeting a particular pathway could collectively be selected for to promote tumour survival. We tested this hypothesis in tumours with hypermutation.

We identified six tumours that had frequent occurrence of hypermutation and that were known to be frequently associated with microsatellite instability (MSI). We identified NMD-sensitive genes (with low z-scores) that were enriched in NMD-elicit mutations in the hypermutated samples when compared to other types of mutations such as silent mutation. This analysis identified genes that were involved in DNA repair such as *MSH2*, *MSH6* and *ASCC3* (Supplementary Data 2; false

discovery rate (FDR) $< 1e - 06$). Moreover, pathway enrichment analysis identified DNA repair as one of the most significantly enriched pathways (Fig. 3a and Supplementary Data 3). In addition, the analysis identified other pathways that appeared to be enriched in NMD-elicit mutations in hypermutated samples. For example, there was a significant enrichment in NMD-elicit mutations in genes involved in chromatin remodelling/ modification, such as *SMARCAD1*, *CHD1*, *CHD8*, *HDAC4*, *BRD3* and *TTF1* (FDR $< 5e - 05$). There was also an enrichment in pathways involved in RNA binding and splicing. Most notably, there were frequent NMD-elicit mutations in genes such as *LARP4B*, *YTHDC1* and *EIF5B* (FDR $< 5e - 05$ and Fig. 3). These results suggest a possible requirement for suppression of translation of mutated genes in hypermutated samples to promote cancer cell survival. Further analysis revealed an unexpected enrichment of NMD-elicit mutations in *LARP4B*

**Table 1 | Top NMD-affected genes in 24 cancers.**

| Cancer type (#samples) | Top genes that frequently harbour NMD-elicit mutations (number of sample with its NMD mutants) |
|---|---|
| Adrenocortical (91) | *TP53** (6), *DCP1A*(5), *MEN1** (5) |
| Bladder urothelial (396) | *KDM6A** (69), *ARID1A** (62), *TP53** (55), *RB1** (48), *STAG2** (39), *TTN*† (31), *EP300** (23), *FAT1** (23), *MUC16*† (23) *CDKN1A* (22) |
| Breast-invasive (982) | *TP53** (93), *MAP3K1** (68), *CDH1** (64) |
| Cervical and endocervical (194) | *MUC4*† (14), *TTN*† (12) |
| Colon (270) | *TTN*† (49), *APC** (45), *RNF43** (31), *TP53** (26), *BMPR2* (20), *ZC3H13* (18), *SYNE1* (17), *ATM** (16), *ARID1A** (14), *MBD6* (14), *MUC16*† (14) *RYR2* (14) |
| Diffuse large B-cell lymphoma (48) | *TNFAIP3** (5), *CIITA** (4), *ARID1A** (3), *HLA-A** (3), *SETD1B* (3), *SPEN** (3), *TET2** (3) |
| Oesophageal (184) | *TP53** (48), *TTN*† (11) |
| Glioblastoma | *ARHGAP5* (50), *NF1** (17) |
| Head and neck squamous cell (512) | *TP53** (114), *FAT1** (86), *NSD1** (42), *CDKN2A** (41), *NOTCH1** (33), *TTN*† (29) |
| Kidney clear cell (213) | *PBRM1** (62), *CDC27* (18), *FAM151A* (17), *OVGP1* (17), *PHACTR1* (17), *VHL** (17), *FANCE** (16) *DNMT1* (15), *GLI1* (15), *MSI1* (14) |
| Brain lower-grade glioma (516) | *ATRX** (45) |
| Liver hepatocellular (373) | *TP53** (32) *APOB* (19) |
| Lung (543) | *TP53** (65), *TTN*† (45), *CSMD3* (33), *MUC16*† (33), *LRP1B* (31), *STK11** (30) *XIRP2* (28) |
| Lung squamous cell (178) | *TP53** (32), *TTN*† (26), *CSMD3* (19), *MUC16*† (12), *FAT1** (11) |
| Ovarian (374) | *TP53** (42) |
| Pancreatic (150) | *TP53** (26), *SMAD4** (13), *CDKN2A** (11) |
| Rectum (116) | *APC** (20), *TTN*† (16), *TP53** (15), *FBXW7** (9) *MAL2* (6), *VPS13C* (6) |
| Sarcoma (259) | *ATRX** (30), *TP53** (24), *RB1** (21) |
| Skin cutaneous melanoma (472) | *TTN*† (102), *NF1** (49), *DNAH5*† (44), *ARID2** (37), *MUC16*† (35), *LRP1B* (34), *DNAH7* (26), *TP53** (26) *DCC* (25), *DNAH8* (24) |
| Stomach (289) | *ARID1A** (57), *RNF43** (56), *TTN*† (55), *TP53** (44) *BZRAP1* (33), *XYLT2* (27), *LARP4B* (23), *MBD6* (23), *ZC3H13* (22), *PLEKHA6* (19) |
| Uterine corpus endometrioid (248) | *PTEN** (96), *TTN*† (66), *ARID1A** (53), *PIK3R1** (44), *CTCF** (30), *MUC16*† (21), *FAT1** (20), *ZFHX3** (20), *DNAH11*† (18) *MKI67* (18) |

NMD, nonsense-mediated decay.
Note: only the top 10 genes most frequently affected by NMD-elicit mutations are shown for each cancer. Three cancers (KIRP, PRAD and THCA) do not have top NMD-affected genes defined in Methods.
*COSMIC-annotated cancer drivers.
†Potential false-positive cancer-associated genes.

(22/54 compared) and *EIF5B* (12/54) in hypermutated stomach adenocarcinoma cases. Note that 25 out of 54 hypermutated cases had NMD-elicit mutations of at least one of these two genes (Fig. 3b). This indicated that suppression of translation was particularly important in hypermutated STADs. Moreover, we observed that 12 out of the 54 (22%) hypermutated STADs also had *PTEN* NMD-elicit mutations. Compared to other cancers, there was a significant enrichment of NMD-elicit mutations in the *PTEN* of hypermutated STADs ($P = 4e - 05$).

Interestingly, 21 out of the hypermutated samples, that also had high MSI, with NMD-elicit mutations in *LARP4B* occurred because of deletion or insertion of a T at position chr10:890939, hg19 in the *LARP4B* gene. As expected, NMD-elicit mutations in *LARP4B* were associated with a modest but significant reduction in expression in STAD (median ratio of REV = 0.15, FDR = 0.017, one-side MWW test). Similarly, 8 out of 12 NMD-elicit mutations in *PTEN* occurred because of a deletion of an A (chr10:89717770-A, hg19). NMD-elicit mutations in *PTEN* were associated with a profound and significant reduction in *PTEN* expression (median ratio of REV = 0.02, FDR = 8.7 e − 05, one-side MWW test). To confirm these results, we analysed data from an independent study on STAD that conducted whole-genome sequencing of 100 STADs[13]. This revealed that 4 out of 10 cancers with MSI had mutations that would induce PTCs in *LARP4B* and three of those mutations occurred at the same site mentioned in TCGA data (chr10:890939-T, hg19). Two out of the ten MSI samples had frameshift mutations in *EIF5B*. For *PTEN*, there were three cancers that had NMD-elicit mutation out of the ten MSI cancers (Fig. 3c). These results indicate that NMD-elicit mutations in these genes play a permissive role and are, therefore, potential therapeutic targets for hypermutated STADs with MSI.

## Discussion

Although the mechanisms and the rules of NMD have been reported[5,9], the full extent of its contribution to cancer initiation and progression has remained unclear. In this work we provide a systematic annotation of TCGA mutations as to whether or not they are likely to induce NMD. We show a global enrichment of NMD-elicit mutations in genes involved in DNA repair, chromatin modifications and RNA stabilization in hypermutated tumours. We speculate that this enrichment plays a permissive role to allow the survival of cancer cells with hypermutation and MSI. Our work suggests that investigations into the therapeutic opportunities that target the dependence of tumours on NMD-mediated LoF could lead to positive outcomes.

Previous annotations of frameshift and NMs do not allow the clear identification of whether or not a mutation is likely to elicit NMD[7,14]. Our analysis used precise prediction of the PTC that results from each of the mutations examined. This enabled a large-scale re-annotation of mutations and revealed that only two-third of nonsense and frameshift mutations can elicit NMD. Our catalogue of NMD-elicit and NMD-escape mutations was validated by demonstrating the significant downregulation of expression in only the predicted NMD-elicit group (Fig. 1d). Interestingly, some in-frame and missense mutations were predicted as NMD-elicit mutations, which suggests that potential PTCs may be overlooked in these types of mutations. We found that an in-frame mutation can result in the approximation of nucleotides that result in a PTC. Similarly, missense mutations occurring in the start codon can result in a late-start codon that is out of the normal frame resulting in a frameshift and, consequently, a PTC (see Methods).

We found that NMD-elicit mutations target certain TSGs (for example, *TP53* and *NF1*) across all cancer types, which is

consistent with previous studies[15–17]. Interestingly, we found that the magnitude of associated reduction in expression is higher in TSGs compared with other genes. In other words, the NMD-elicit mutations are associated with a more significant decline in the expression of TSGs. One possible explanation is that TSGs, as we

observed, tend to have one allele deleted and the other one harbouring NMD-elicit mutations (Supplementary Fig. 3a), which is in line with the two-hit hypothesis for TSGs[18]. However, whether, on a global scale, NMD-elicit mutations occur more frequently than other LoF mutations is difficult to

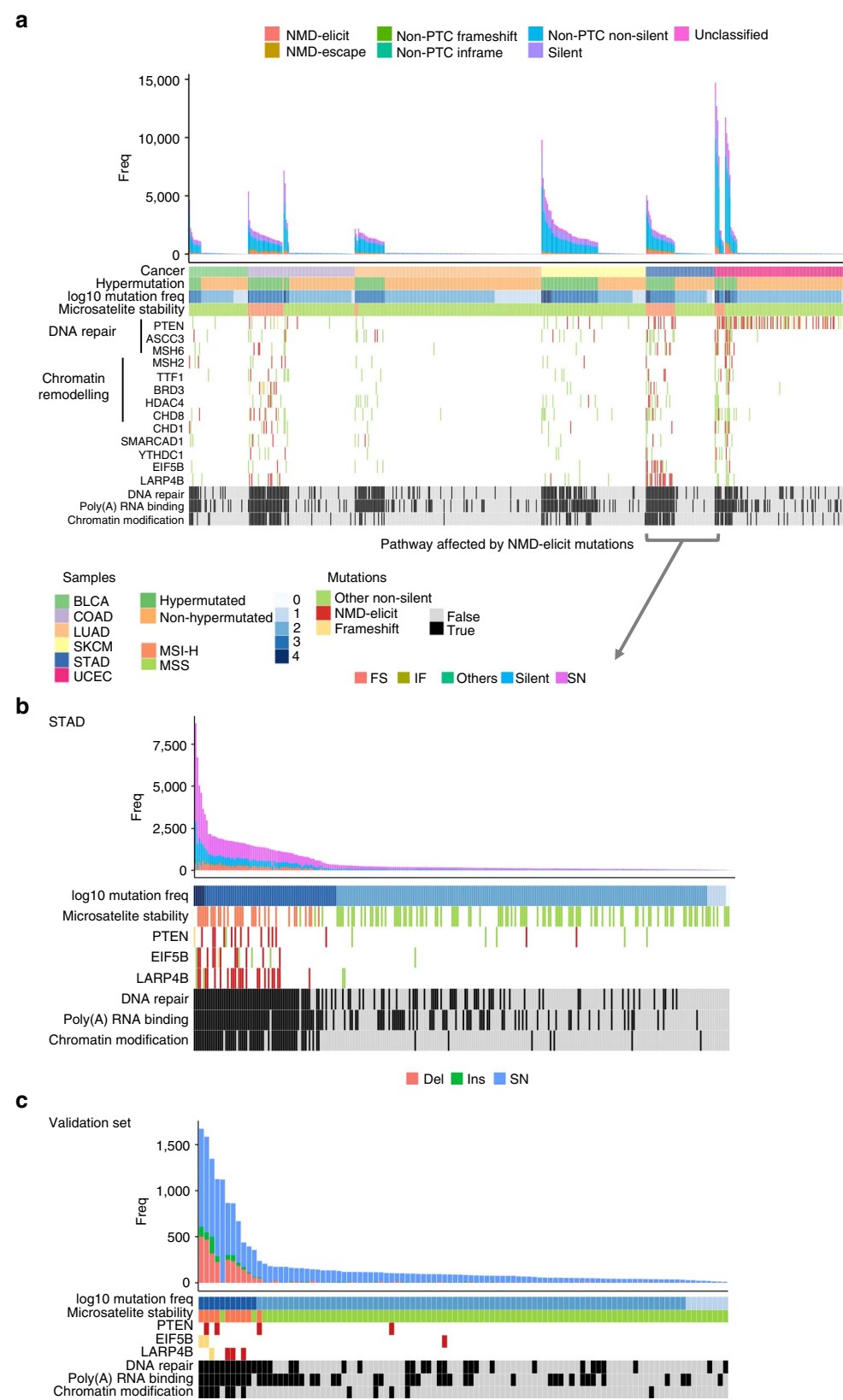

predict. This is because it is difficult to conclude whether or not a non-synonymous mutation results in LoF without conducting functional studies. We speculate that NMD-elicit mutations would occur less frequently than other LoF mutations in some genes such as *TP53*, but that they are associated with significant LoF because of reduction in expression particularly in TSGs. The association between NMD-elicit mutations in TSGs and the very low expression level further supports the viewpoint that NMD is a promising target for cancer treatment. Recent work into the mechanisms of NMD regulation has provided feasible approaches[4,19–21] for the inhibition of NMD[22–25]. The aim of all these approaches is to restore the expression of full-length genes that harbour NMD-elicit mutations[26] by 'reading through' the PTC. We also found evidence of tumour type-specific targeting of driver TSGs by NMD (Table 1), suggesting that therapeutic targeting of NMD could be successful in a wide range of tumours. In addition, inhibition of NMD can result in the expression of new antigenic epitopes and enhanced antitumour immune response[27]. However, other complexities need to be considered. For example, the read-through product that is generated from a PTC-harbouring transcript may result in a negative-dominant effect or other deleterious impacts. It is noteworthy that certain tumours (UCEC, BLCA and STAD) appear to have a higher load of NMD-elicit mutations, suggesting that they may be particularly sensitive to therapeutic targeting of NMD. Whether this higher load of NMD-elicit mutations is the result of specific mutational signatures that occur in these particular tumours is testable in future studies. In addition, further analysis of the impact of cancer-specific mutational partterns[28] and signatures on potential enrichment of NMD-elicit mutations in particular pathways might provide important insights into the possible dependencies of these tumours on NMD.

We observed that NMD-elicit mutations in 'cancer-unrelated' genes caused a modest reduction in expression but clustered in particular pathways presumably to permit the emergence of certain tumour phenotypes. This is reminiscent to the concept of BRCAness[29], where the dysfunction of many different genes can perturb homologous recombination repair in a relatively large proportion of patients. We found that DNA repair, chromatin remodelling and RNA-binding pathways are preferentially targeted by NMD-elicit mutations in hypermutated samples. Given the known role that DNA repair and chromatin remodelling genes play in regulating response to DNA damage, it is plausible to suggest that this enrichment acts in a permissive capacity to enable the establishment and survival of hypermutated cancer cells[30]. Moreover, we found that stomach adenocarcinoma (STAD) tends to have disproportionately higher number of NMD-elicit mutations and a mutator phenotype that selects for NMD-elicit mutations in *LARP4B*, *EIF5B* and *PTEN*. Although the potential relationship between *LARP4B* or *EIF5B* and cancer has been previously reported[31–34], the link between

these genes and hypermutation was previously unrecognized. It is conceivable that NMD-mediated hypofunction of *LARP4B* or *EIF5B* permits the hypermutated cancer cells to cope with a high mutation load by delaying translation.

Although the rules that we used to predict NMD-elicit mutation are known to have a significant effect on mRNA levels, other potentially important factors may have been overlooked because of the incomplete understanding of the mechanisms of NMD in humans. For example, the potential role of insertions and deletions in the 3′ untranslated repeat (UTR) in inducing NMD has remained unclear. Whether or not mutations in members of the NMD complex may have an impact on eliciting NMD is difficult to study on a global scale because of the low number of mutations in this complex and the lack of clarity as to whether any particular mutation in this complex is associated with LoF. In addition, our results do not take into account the potential suppression of NMD by the tumour microenvironment as recently reported[35,36]. Other than NMD-elicit mutations, mutations occurring in transcription factors, epigenetic factors and non-coding mRNAs may affect mRNA levels. Another unaddressed question is the relationship between the mRNA level and the combination of allelic ratio, allele-specific expression and the prevalence of a mutation. The future application of advanced sequencing technologies, and the development of novel computational models would enable further detailed analyses of the effect of a mutation on gene expression.

In summary, we provide a comprehensive annotation of NMD in all mutations reported by TCGA. Our work unravels unrecognized dependencies of tumours on NMD-mediated LoF with potential therapeutic opportunities[37–41].

## Methods

**Data pre-processing.** The following four files, RNA-seq data (tcga_RSEM_Hugo_norm_count, 2016-02-18), somatic mutations annotation file (MAF, 2016-04-28) for results of DNA sequencing, CNV data (Gistic2_CopyNumber_Gistic2_all_-thresholded.by_genes, 2016-08-16) and clinical data (PANCAN_clinicalMatrix, 2016-04-30, were downloaded from the 'TCGA Pan-Cancer (PANCAN)' cohort at the TGCA hub and the GA4GH-BD2K (TOIL) hub on USCS Xena (https://xenabrowser.net)[42]. The TCGA Pan-cancer RNA-seq data contained normalized and log-transferred counts, which were quantified by RSEM[43]. We downloaded the non-negative matrix factorization (NMF) cluster data (Version 2016_01_28) from the Broad GDAC Firehose (http://gdac.broadinstitute.org)[44], which clustered samples based on mRNA-seq data. To obtain sufficient samples for adequate statistical analysis, we only analysed the 24 cancer types that had more than 10,000 somatic mutations in total.

**Prediction algorithm of NMD-elicit mutations.** We retrieved the coding sequences (CDSs) and positional annotations of transcripts provided by UCSC (hg19, Feb. 2009) via the R interface. The canonical isoform was defined as the one with the longest CDS. For each gene, we extracted the positional information from the TxDb package and the longest CDS from the BSgenome package, based on the reference genome (hg19/NCBI Build 37). The positional information included the chromosomal start and stop positions of exons. We calculated the relative positions of exon–exon junctions and the relative position of mutations on the CDSs.

**Figure 3 | Hypermutated tumours are enriched in NMD-elicit mutations targeting hypermutation-permissive pathways.** (**a**) The barplot in the top panel shows the numbers of the different types of mutations (*y* axis) in each sample (*x* axis). The heatmap in the bottom panel shows the cancer type (cancer), whether a sample is hypermutated or not (hypermutation), log 10 of total mutation numbers (log 10 mutation frequency), whether a sample is microsatellite-stable or not (microsatellite stability), the NMD-elicit mutations in individual relevant genes and their pathway affiliation as indicated and the pathways affected by NMD-elicit mutations (black means that the pathway has at least one gene harbouring NMD-elicit mutations) in each sample (*x* axis). (**b**) In STAD, the heatmap shows the log 10 of total mutation numbers (first row), MSI (second row), the NMD-elicit mutations in individual relevant genes (3rd–7th rows) and NMD-elicit mutations in genes that belong to certain pathways (8th–10th rows) in each sample (*x* axis). FS, frameshift indels; IF, in-frame indels; SN, single-nucleotide substitution mutations. (**c**) In gastric cancer validation cohort[13], the heatmap in the bottom panel shows the log 10 of total mutation numbers (first row), MSI (second row), the NMD-elicit mutations in individual relevant genes (3rd–7th rows) and NMD-elicit mutations in genes that belong to certain pathways (8th–10th rows) in each sample (*x* axis). FS, frameshift indels; IF, in-frame indels; SN, single-nucleotide mutations. Del, deletions; Ins, insertion; SN, substitution mutations. Note that the NMD-escape mutations in *EIF5B* and *LARP4B* are likely to induce NMD as they failed the NMD rules only marginally. Both *EIF5B* NMD-escape mutations failed the rules because they generate PTCs that are 175 bp away from the start codon. The *LARP4B* mutations generate a PTC that is 187 bp away from the start codon.

Combining the mutation data from TCGA and the wild-type CDSs gave the mutated sequences. For this analysis, we have excluded 22 genes with PTCs in the wild-type sequence, mutations that occur on the splice sites or on the exon–intron junctions not annotated by TCGA, mutations where the reference allele does not match the nucleic acid on the reference genome, 32 genes with CDS that do not start with start codons, which may be incorrectly annotated, and 47 genes starting with start codon but without in-frame stop codon. The reason for the first criteria is that the mRNA may have strong second structure where the NMD fails. The reason for the second criteria is that the splice-site mutations may induce alternative splicing, which affects mRNA levels in a complex way. The third criterion is to exclude the possible incorrect mutation annotation.

Our algorithm first detected all possible start codons (ATA and ATG) and stop codons (TGA, TAA and TAG) in their mutated sequences. Among all of the possible start codons, the most upstream one was denoted as the putative start position. The stop position was selected by the two criteria. The first one is that the stop codon is in the same frame as the start codon, because the CDS must be the multiples of three. The second one is that the stop position is the closest one to the start codon among all the candidates.

On the basis of the predicted relative positions of the stop codon and the last exon–exon junction in the variant, the classification of NMD was conducted by three rules, that the gene has at least two exons, that the stop codon in the variant is more than 50 bp upstream from the last exon–exon junction and that the PTC is more than 200 bp downstream from the start codon. If the mutation satisfied the above rules, the mutation was classified into the NMD-elicit mutation; otherwise, into NMD-escape mutations that generate PTCs, or non-PTC mutations.

In addition to the aforementioned filtering, the certain types of mutations were also excluded from analysis. The first type is the mutations of genes without Entrez gene IDs or with multiple corresponding Entrez gene IDs. The second type is the repeated annotations of the same mutations, but multiple mutations within one gene in one sample are retained to avoid underestimation of NMD effect. The third type includes the mutations with TCGA annotation as 'Splice_Site', '3′UTR', '5′UTR', 'IGR', 'Intron' and 'RNA'.

### Novel annotation compared to TCGA.
We detected NMD-elicit mutations from the missense mutations, NMs, silent mutations, nonstop mutations, mutations on the translation start site, in-frame indels and frameshift indels annotated by TCGA. We compared our prediction results (NMD-elicit mutations versus others) with the original TCGA annotations (Fig. 1c). We validated unexpected results, such as reclassifying in-frame indels or missense mutations to NMD-elicit mutations by performing manual checks. Supplementary Fig. 4 gives examples of how in-frame indels or missense mutations can potentially elicit NMD. We measured the correlation between total mutations and NMD-elicit mutations by Pearson correlation and identified the cancers with higher NMD-elicit mutated rate by generalized linear regression on cancer types.

### Expression-based analysis on NMD impact.
*Statistical test.* We compared the expression levels using the nonparametric MWW test (wilcox.test in R) or the *t*-test. The MWW tests and *t*-tests without the statement of one-sided are two-sided.

*Calculation of REV.* To validate the impact of NMD-elicit mutations on the expression levels, we calculated the relative expression of variant to mask the gene-specific effect for each mutation, including NMD-elicit and other mutations. The relative expression was defined by the rank of the expression level of the variant (the gene with a mutation) in its background expression. The gene-and-cancer-specific background expression was defined as the expression levels of a gene from these samples from the same cancer, which have no somatic mutations or CNV and belong to the same NMF cluster (Broad firehose) or from the same cancer if the NMF clustering was absent. For example, in $J$ total samples, gene $i$ has NMD-elicit mutations in $m$ samples ($1 \sim m$), and does not have NMD-elicit mutations in the other $n$ samples ($m+1 \sim m+n$). Denote $e_{i,j}$ as the expression level of gene $i$ in sample $j$, $\mathbf{E} = e_{i,m+n+1}, e_{i,m+2+2}, e_{i,j}$ as the background expression. The REV of each mutation of gene $i$ in sample $m$ can be calculated by $REV_{i,m} = \frac{R_{i,m}-1}{size(\mathbf{E})}$, $size(\mathbf{E}) = J - m - n$, where the $size(\mathbf{E})$ denotes the number of wild-type samples from the same cancer and the $R_{i,m}$ denotes the ranking of $e_{i,m}$ in the background expression level. The lower the relative expressions of variant, the larger the downregulation of expression of the variant. We compared the relative expression of predicted NMD-elicit mutations with the NMD-escape or non-PTC mutations in the frameshift deletions, frameshift insertions and NMs, by the one-side MWW test (wilcox.test in R).

### Annotation of TSGs and cancer-specific NMD signature.
The list of cancer-related genes and TSGs was downloaded from COSMIC (http://cancer.sanger.ac.uk/census)[12]. The number of studied TSGs is 71. The 127 SMGs were reported by pan-cancer mutation analysis[11]. We defined the top NMD-affected genes for each cancer as the genes harbouring NMD mutations within over 5% samples. We marketed genes that are potential false-positive cancer-associated genes, such as *TTN* and *MUC16* (ref. 28; Table 1).

### Quantifying gene-specific NMD-associated reduced expression.
To mask the NMD-insensitive genes that do not show decrease in their expression even with

NMD-elicit mutations, we defined and calculated the *z*-score based on the $U$ statistic in the nonparametric MWW test. Every gene in each cancer had its gene-specific and cancer-specific *z*-score. For each gene, we extracted the expression levels of its NMD-elicit mutants and the levels of its predefined wild type. To calculate the valid statistic, we discarded the gene with less than three NMD-elicit mutations. We denoted $N_1$ as the number of samples with the wild type of gene, and $N_2$ as the number of samples with NMD-elicit mutations in this gene. We ranked all the expression levels in two groups and calculated the sum of ranks (denoted as U1 and U2) for each group separately. The *z*-score is the statistic $U$ normalized by $N_1 \cdot N_2$, which ranges from 0 to 1.

$$U_1 = \text{sum rank} - \frac{N_1(N_1+1)}{2}$$
$$U_2 = \text{sum rank} - \frac{N_2(N_2+1)}{2}$$
$$U = \min(U_1, U_2)$$

The lower *z*-score means the more decreased expression of the genes that harbours NMD-elicit mutations. The *z*-scores no less than 0.5 means the magnitude of NMD-associated reduction in expression is negligible in this gene. On the basis of *z*-scores, we classified genes into NMD-sensitive genes (*z*-scores $\leq 0.3$), intermediate ($0.3 < z$-scores $\leq 0.4$) and NMD-insensitive ones (*z*-scores $> 0.4$),

### NMD-elicit mutations in hypermutation.
We defined hypermutated samples as the ones with over 1,000 filtered mutations. The non-hypermutated samples were the ones with less than 100 filtered mutations. BLCA, COAD, LUAD, SKCM, STAD and UCEC had more than 30 hypermutated samples and were used for further analysis. The MSI data were downloaded from previous pan-cancer analysis[45].

To quantify the gene-specific enrichment in NMD-elicit mutations in hypermutated samples, we calculated the normalized ratio between NMD-elicit mutations and silent mutations ('Silent', '3′ UTR', '5′ UTR', 'Intron', 'RNA') between the hypermutated and non-hypermutated samples. We calculated the $P$ value by Fisher exact test, adjusted by FDR. The ratio can measure the enrichment of NMD in hypermutation. It was normalized by the total number of each mutation type to mask the negative selection on the NMD-elicit mutations. To identify important pathways or molecular functions, we took the significantly enriched NMD-sensitive genes (FDR $< 0.05$) as the input for the gene ontology enrichment analysis (top GO in R) by the classic and weighted algorithm. The validation data set was from an independent data set[13] other than the TCGA data.

### Data availability.
The pan-cancer data and the computer codes that support the findings of this study are available in GitHub (https://github.com/ZYBunnyHu/NMD-paper) without restriction. The R package for predicting NMD-elicit mutations, masonmd (Make Sense of NMD), are available in GitHub with the identifier (DOI: 10.5281/zenodo.546698)[46].

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

## Acknowledgements

This work was supported by the Medical Research Council (Ref: G0902418), Ovarian Cancer Action (Ref: HER00070) and the Oxford Biomedical Research Centre, National Institute of Health Research (Ref: IS-BRC-0211-10025). Z.H. acknowledges the support of the China Scholarship Council—Nuffield Department of Medicine Scholarships (Ref: GAF1516_CSCUO_839316) from the China Scholarship Council (on behalf of the Chinese Ministry of Education) and the University of Oxford. C.Y. acknowledges the support of an UK Medical Research Council New Investigator Research Grant (Ref: MR/L001411/1) and a Wellcome Trust Core Award (Ref: 090532/Z/09/Z).

## Author contributions

Z.H., C.Y. and A.A.A. wrote the manuscript. Z.H., C.Y. and A.A.A. conducted the analyses. C.Y. and A.A.A. conceived and supervised the research.

## Additional information

**Competing interests:** The authors declare no competing financial interests.

