## [Peer Review File · Nature Communications]

Reviewers' Comments:

Reviewer #1 (Remarks to the Author)

In this study, the authors investigate the effect of nonsense mediated decay (NMD) on gene expression. Based on their classification of NMD mutations, they find that these are strongly correlated to reduced gene expression. This effect is stronger for known tumor suppressor genes. For hypermutated samples, the authors observe preferential occurrence of NMD for genes involved in essential cell functions like chromatin remodeling and translation. An in-depth analysis of STAD shows that NMDs preferentially occur in two translational initiators. Generally, this is an interesting study, but as described below missing a) putting NMDs in relation to other LoF variation and b) a more rigid statistical approach to exclude confounding factors on the analysis.

General points for the manuscript:

1) When reporting significant (down)regulation, it would be helpful to report the actual change. Any difference will be significant given large enough sample size.

While the analysis results seem statistically sensible, the following point should be made more clear in the methods:

2) When calculating change in expression, do the authors only consider the primary transcript in their analysis? Otherwise, NMD could simply lead to shorter transcripts, which would be aligned to an alternative transcript and thus only would seem to be reduced

Regarding the statements and comparisons in the manuscript, I am missing a general baseline of variations in the analyzed samples:

3) TSGs harbor more premature termination codon: How does that number is different from other loss-of-function (LoF) mutations?

4) NMD-elicited vs. NMD-escape: Later in the manuscript, it is stated that NMD-elicited

co-occur with the deletion of the other allele. How does this comparison change when removing these cases?

5) When stating that three cancer types show higher numbers of NMD-elicited mutations: Is that affected by the difference in mutation patterns these cancer types exhibit? (see Lawrence et. Al, "Mutational heterogeneity in cancer and the search for newcancer-associated genes")

6) When reporting that tumor suppressor genes are more often affected than expected by chance, is it at the same rate as other LoF mutations?

7) As in comment 2, more generally: How do the authors control for deletion of the other allele of an affected gene? If NMD has an effect, it should be possible to see a difference between no NMD, NMD only and NMD+deletion of the other allele.

8) As in comment 4, more generally: Are NMDs significantly different in frequency than other LoF variants, especially considering hypermutated samples? Or would one get the same results when looking for LoF variants?

9) For the analysis of hypermutation in stomach adenocarcinoma: Although the authors report a P value in the following paragraph, it is not clear how big the change in expression was and if there was a corresponding deletion of the alternative allele.

10) When observing similar mutation/insertion/deletion for a gene: How likely is this change based on sequence context, e.g. mutational preference of the tumor type and possibility of actual introducing a new stop codon at a random position.

11) The authors predict NMD-elicited "in frame" and missense mutations, but not much detail is provided. What is the frequency and gene distribution of these predicted events? Are they enriched in TSGs?

12) Finally, the speculation in the discussion that NMD blockade might be useful therapeutically is rather vague. Most rescued transcripts would still produce defective proteins. Could the potential rescue frequency be quantified?

Reviewer #2 (Remarks to the Author)

„A pan-cancer genome-wide analysis reveals tumour dependencies by induction of nonsense mediated decay“.

The paper is touching on an important subject , predicted loss of function mutation and nonsense mediated decay.

Overall. Not seeing the author name is ok. However, by judging by the number of unclear facts, the manuscript would benefit by having either more authors going over the manuscript.

1-In their treatment of the question, the author(s) remains unclear sometime talking about causation, sometimes talking about association.

2-The study is based on the pre-existing TCGA dataset, but almost no explanation of what the author used is provided.

3-The authors mention 24 cancer types but TCGA has a number of other cancer they study, the author(s) should mention how the selection has been done.

4-The data used from TCGA are mainly the DNA and RNA sequencing experiment but this is almost never mentioned.

5-Also, the majority of these mutations are likely to be somatic in origin and this is not much discussed. Somatic mutations can come with different allelic ratio in the tumour sample and this has specific consequences.

6- Page 1: TSG should be defined in full before the acronym is used

7-Page 2 Ch 2: the authors mention a validation; they should spell out if it an experimental one

8-Page 3 Line 6: The author mention „unexpectedly“ but they should mention if they expected more or less.

9-Page 1 line 7 when mentioning that 6% of the million mutations are NMD-elicited. I am not sure this number is relevant

10-Page 3 chapter 2. When the author mention expression level, they mention lower expression.

The authors should clarify if these if they mean it in an absolute or relative

manner.

It is clear from the figure that it is relative but the sentence is ambiguous.

11- Page 3 Chapter line 6

When test on stomach and kidney are mentioned the authors quote P value threshold of 0.007. It is not clear if the authors are taken into account multiple testing.

12- Page 3 last sentence of chapter 2.

It is not clear to me that the result presented are indicating a contribution of loss of function.

It could just be a „passenger“ event at this stage of the evidence presented.

13- Page 3 chapter 3- the number of studied TSG genes should be quoted

In the same chapter the author mentioned: „mutations resulted in a significant reduction in gene expression“. There is no evidence that the lower expression is a consequence of the mutations. It is just an association. And should be described as such

14- Page 4 line 5: 29%. IS it out of all n=XXX samples ? In that case the number should be given.

15- Page 4 chapter 2. The authors mention the occurrence of mutations and deletion. The author should describe how large of a deletion they could detect.

Then when they perform an assessment of expression, should they look at allele specific expression or at least discuss the limitation if they cannot.

16- Spell out MWW when first mentioned

17- Table 1: the acronym of Cancer types is not easy for reader. Mention organ.

18- Author mentioned, „ profound reduction in gene expression at TSG is most probably the result of the mutated allele and deletion of the wild type allele“. Can they assess it by allele specific expression?

19- Discussion: number and overview in early stage of discussion should also be present in the Introduction or early part of the results.

20- Figure 1: The use of panel is a bit excessive in panel B of figure 2, nothing is readable

21- Page 7 chapter 2 sentence 2. The author talk about efficiency that directly imply causation when talking about TSG, whereas it is clearly stated that the two hits hypothesis with co-occurrence of deletion is a possible explanation.

I suggest removing the term NMD efficiency

22- In table 1: we note that a lot of gene such as Titin and Mucin, correspond to some of the biggest and most polymorphic gene in the genome. Would correcting by gene size necessary?

Reviewer #3 (Remarks to the Author)

This manuscript assesses the prevalence and impact of NMD-provoking mutations in the TCGA data set. These authors use a previously derived algorithm to identify ~75,000 NMD provoking mutations, and assess how affect RNA expression from the TCGA data, and their enrichment in tumor suppressor genes, frequently mutated genes in cancer, and amongst distinct cancers.

This is an interesting paper and will be of interest to those studying NMD and those in the cancer field. However it suffers from some deficiencies inherent in their methodology.

1. Their rule for identifying NMD provoking mutations is likely not complete nor as sensitive or specific as they would like. Although they did find decreased mRNA expression in those transcripts that fit their rule in general, it is likely that other mutations, deletions/insertions in the 3'UTR also lead to NMD, and that some of these mutations identified as NMD provoking are not degraded by NMD.

2. It is unclear the contribution of mutations in members of the NMD complex (described by the Wilkinson and recently the Steitz groups) and/or suppression of NMD by the tumor microenvironment (described by the Gardner group) may impact their findings.

3. The role of mutations in the promoter, transcription factors, epigenetic factors, and/or non-coding mRNAs likely has a great impact on mRNA expression, and is difficult to take into consideration.

4. While interesting, it is difficult to know the significance of differences found in tumor types, and perhaps this could be better addressed in the discussion.

Lawrence Gardner

Response to reviewers' comments

Reviewer #1 (Remarks to the Author):

“In this study, the authors investigate the effect of nonsense mediated decay (NMD) on gene expression. Based on their classification of NMD mutations, they find that these are strongly correlated to reduced gene expression. This effect is stronger for known tumor suppressor genes. For hypermutated samples, the authors observe preferential occurrence of NMD for genes involved in essential cell functions like chromatin remodeling and translation. An in-depth analysis of STAD shows that NMDs preferentially occur in two translational initiators. Generally, this is an interesting study, but as described below missing a) putting NMDs in relation to other LoF variation and b) a more rigid statistical approach to exclude confounding factors on the analysis.”

“1) When reporting significant (down)regulation, it would be helpful to report the actual change. Any difference will be significant given large enough sample size.”

We thank the reviewer for the suggestion. We have added the ratio of the medians to all two-group comparisons.

“2) *While the analysis results seem statistically sensible, the following point should be made more clear in the methods: When calculating change in expression, do the authors only consider the primary transcript in their analysis? Otherwise, NMD could simply lead to shorter transcripts, which would be aligned to an alternative transcript and thus only would seem to be reduced*”

- a- Yes, we only considered the primary transcript.
- b- Please note that a premature stop codon would not be expected to result in truncating the transcript but would be expected to truncate the protein. This is because such stop codon signals “no” amino acids and therefore, translation would be expected to stop. The stop codon does not normally signal for stopping the RNA polymerase.

“3) Regarding the statements and comparisons in the manuscript, I am missing a general baseline of variations in the analyzed samples: TSGs harbor more premature termination codon: How does that number is different from other loss-of-function (LoF) mutations?”

- a- We do not claim that NMD-elicited mutations occur more frequently than other LoF mutations in TSGs. Our findings suggest that NMD-elicited mutations occur more frequently in TSGs and Cancer related genes compared to other genes.
- b- The comparison that is suggested by the reviewer is difficult to make because it is, perhaps, impossible to classify a non-synonymous mutation into LoF or not without

doing functional studies on all reported non-synonymous mutations. However, to address this point, we have added the following sentences in discussion:

“Whether, on a global scale, NMD-elicited mutations occur more frequently than other loss-of-function (LoF) mutations is difficult to predict. This is because it is difficult to conclude whether or not a non-synonymous mutation results in LoF without conducting functional studies.”

“4) NMD-elicited vs. NMD-escape: Later in the manuscript, it is stated that NMD-elicited co-occur with the deletion of the other allele. How does this comparison change when removing these cases?”

Thank you for raising this point. We compared the relationship between NMD-elicited mutations and the expression levels of TSGs in the presence or absence of a deletion. We observed that NMD-elicited mutations are associated with a significant reduction of gene expression when compared to other types of mutations independent from whether or not there is an associated deletion of an allele, see the figure below. We have now added this comparison as supplementary figure 3b. We also added the following text in the results section:

“Importantly, in the absence of a deletion of an allele, NMD-elicited mutations are still associated with a significant reduction of gene expression but at a lower magnitude compared to cases when a deletion is present ($P < 2.2e-16$, MWW test, fold change = 0.43 and $P < 2.2e-16$, MWW test, fold change = 0.09, respectively, Supplementary Fig. 3b).”

“5) When stating that three cancer types show higher numbers of NMD-elicited mutations: Is that affected by the difference in mutation patterns these cancer types exhibit? (see Lawrence et. Al, “Mutational heterogeneity in cancer and the search for new cancer-associated genes”)”

This is a very interesting question but perhaps beyond the scope of our current manuscript. The hypothesis that certain mutational patterns that are specific to a particular tumour type could

be associated with higher number of NMD-elicited mutations or perhaps NMD-elicited mutations that target particular pathways is testable. To address this comment, we have added the following in the discussion:

“Whether this higher load of NMD-elicited mutations is the result of specific mutational signatures that occur in these particular tumours is testable in future studies. In addition, further analysis of the impact of cancer-specific mutational patterns²⁷ and signatures on potential enrichment of NMD-elicited mutations in particular pathways might provide important insights into the potential dependencies of these tumours on NMD.”

“6) When reporting that tumor suppressor genes are more often affected than expected by chance, is it at the same rate as other LoF mutations?”

- a- We speculate that NMD-elicited mutations would occur less frequently compared to other LoF mutations in some genes. For example, the majority of TP53 missense non-synonymous mutations are thought to be LoF mutations. These occur in (22.2% of 7725 cancer samples) compared to NMD-elicited mutations (8.8% of 7725 cancer samples). To address this point, we added the following in the Results section:

“However, the frequency of NMD-elicited mutations appears to be lower than other loss-of-function mutations. For example, NMD-elicited TP53 mutations occurred in 8.8% of all cancer samples (n=7725) while missense non-synonymous mutations in the same gene occurred in 22.2% of samples.”

- b- We also added the following in the discussion section:

“It is important to consider NMD-elicited mutations in other loss-of-function (LoF) mutations. However, it is difficult to predict LoF mutations in silico without functional studies. We speculate that NMD-elicited mutations would occur less frequently than other LoF mutations in some genes such as TP53 but that they are associated with significant loss of function due to reduction in expression particularly in TSGs.”

“7) As in comment 2, more generally: How do the authors control for deletion of the other allele of an affected gene? If NMD has an effect, it should be possible to see a difference between no NMD, NMD only and NMD+deletion of the other allele.”

This has been addressed under point number 2.

“8) As in comment 4, more generally: Are NMDs significantly different in frequency than other LoF variants, especially considering hypermutated samples? Or would one get the same results when looking for LoF variants?”

We observed that the frequency of NMD-elicited mutations correlated well with the frequency of mutations in general. Please see figure 1e.

“9) For the analysis of hypermutation in stomach adenocarcinoma: Although the authors report a P value in the following paragraph, it is not clear how big the change in expression was and if there was a corresponding deletion of the alternative allele.”

The fold change in expression has now been added.

“10) When observing similar mutation/insertion/deletion for a gene: How likely is this change based on sequence context, e.g. mutational preference of the tumor type and possibility of actual introducing a new stop codon at a random position.”

This has now been addressed under point number 5.

“11) The authors predict NMD-elicited “in frame” and missense mutations, but not much detail is provided. What is the frequency and gene distribution of these predicted events? Are they enriched in TSGs?”

- a- Please note that we have reported the frequency of these events in figure 1c. They are not frequent and, therefore, it is difficult to address the question of enrichment. However, our systematic approach to testing whether a mutation may or may not lead to NMD resulted in the observation of these unexpected events.
- b- To clarify how this could occur, the following sentences have been added in the discussion:
“We found that an inframe mutation can result in the approximation of nucleotides that result in a PTC. Similarly, missense mutations occurring in the start codon can result in a late start codon that is out of the normal frame resulting in a frame shift and, consequently, a PTC.”
- c- The following two examples explain how some inframe mutations and missense mutations can generate premature stop codons and elicit NMD. The Query 1 is mutated coding sequence and the Sbjct 1 is unmutated. We have added these examples in supplementary figure 4.

NMD-elicited inframe deletion:

This example is a deletion of CTG in sample “TCGA-BP-4782-01” at chr16:31213519-31213521 (*PYCARD*). The deletion results in the combination of adjacent T and GA, which is a premature stop codon at 277; this gene has 3 exons and the position of last exon-exon junction is 328.

```

Query 1 ATGGGGCGCGCGCGACGCCATCCTGGATGCGCTGGAGAACC TGACCGCCGAGGAGCTC 60
Sbjct 1 ATGGGGCGCGCGCGCGACGCCATCCTGGATGCGCTGGAGAACC TGACCGCCGAGGAGCTC 60

Query 61 AAGAAGTTCAAGCTGAAGCTGCTGTCGGTGCCGCTGCGCGAGGGCTACGGGCGCATCCCG 120
Sbjct 61 AAGAAGTTCAAGCTGAAGCTGCTGTCGGTGCCGCTGCGCGAGGGCTACGGGCGCATCCCG 120

Query 121 CGGGGCGCGCTGCTGTCCATGGACGCCCTTGGACCTCACCGACAAGCTGGTCAGCTTCTAC 180
Sbjct 121 CGGGGCGCGCTGCTGTCCATGGACGCCCTTGGACCTCACCGACAAGCTGGTCAGCTTCTAC 180

Query 181 CTGGAGACCTACGGCGCCGAGCTCACCGCTAACGTGCTGCGCGACATGGGCCTGCAGGAG 240
Sbjct 181 CTGGAGACCTACGGCGCCGAGCTCACCGCTAACGTGCTGCGCGACATGGGCCTGCAGGAG 240

Query 241 ATGGCCGGGCAGCTGCAGGCGGCCACGCACCAGGGCT---GAGCCGCGCCAGCTGGGATC 297
Sbjct 241 ATGGCCGGGCAGCTGCAGGCGGCCACGCACCAGGGCTCTGGAGCCGCGCCAGCTGGGATC 300

Query 298 CAGGCCCTCCTCAGTCGGCAGCCAAGCCAGGCCTGCAC TTTATAGACCAGCACCGGGCT 357
Sbjct 301 CAGGCCCTCCTCAGTCGGCAGCCAAGCCAGGCCTGCAC TTTATAGACCAGCACCGGGCT 360

Query 358 GCGCTTATCGCGAGGGTCACAAACGTTGAGTGGCTGCTGGATGCTCTGTACGGGAAGGTC 417
Sbjct 361 GCGCTTATCGCGAGGGTCACAAACGTTGAGTGGCTGCTGGATGCTCTGTACGGGAAGGTC 420

Query 418 CTGACGGATGAGCAGTACCAGGCAGTGCGGGCCGAGCCACCAACCAAGCAAGATGCCG 477
Sbjct 421 CTGACGGATGAGCAGTACCAGGCAGTGCGGGCCGAGCCACCAACCAAGCAAGATGCCG 480

Query 478 AAGCTCTTCAGTTTCACACCAGCCTGGAACCTGGACCTGCAAGGACTTGCTCCTCCAGGCC 537
Sbjct 481 AAGCTCTTCAGTTTCACACCAGCCTGGAACCTGGACCTGCAAGGACTTGCTCCTCCAGGCC 540

Query 538 CTAAGGGAGTCCCAGTCTACCTGGTGGAGGACCTGGAGCGGAGCTGA 585
Sbjct 541 CTAAGGGAGTCCCAGTCTACCTGGTGGAGGACCTGGAGCGGAGCTGA 588

```

NMD-elicited missense mutations:

The example is an A to G mutation in sample “TCGA-25-2400-01: at Chr6: 35436212-35436212 in gene *RPL10A*. The mutation changes the original start codon from ATG to GTG. The next start codon is in position 122 and is in a different frame from the original one. This results in a premature stop codon at 269. This gene has 6 exons and the last exon-exon junction is at 483. This is an example of how a missense mutation in the start codon results in a frame-shift that is potentially capable of eliciting NMD

```
Query 1 GTGAGCAGCAAAGTCTCTCGCGACACCCTGTACGAGGCGGTGCGGGAAGTCTGCACGGGA 61
Sbjct 1 ATGAGCAGCAAAGTCTCTCGCGACACCCTGTACGAGGCGGTGCGGGAAGTCTGCACGGGA 61
Query 62 ACCAGCGCAAGCGCCGCAAGTTCCTGGAGACGGTGGAGTTGCAGATCAGCTTGAAGAAGT 121
Sbjct 62 ACCAGCGCAAGCGCCGCAAGTTCCTGGAGACGGTGGAGTTGCAGATCAGCTTGAAGAAGT 121
Query 122 ATGATCCCCAGAAGGACAAGCGCTTCTCGGGCACCGTCAGGCTTAAGTCCAATCCCCGCC 181
Sbjct 122 ATGATCCCCAGAAGGACAAGCGCTTCTCGGGCACCGTCAGGCTTAAGTCCAATCCCCGCC 181
Query 182 CTAAGTTCCTCTGTGTGTCTCTGGGGACCAGCAGCACTGTGACGAGGCTAAGGCCGTGG 241
Sbjct 182 CTAAGTTCCTCTGTGTGTCTCTGGGGACCAGCAGCACTGTGACGAGGCTAAGGCCGTGG 241
Query 242 ATATCCCCACATGGACATCGAGGCGCTGAATAAACTCAACAAGAATAAAAACTGGTCA 301
Sbjct 242 ATATCCCCACATGGACATCGAGGCGCTGAATAAACTCAACAAGAATAAAAACTGGTCA 301
Query 302 AGAAGCTGGCCAAGAAGTATGATGCGTTTTTGGCCTCAGAGTCTCTGATCAAGCAGATTC 361
Sbjct 302 AGAAGCTGGCCAAGAAGTATGATGCGTTTTTGGCCTCAGAGTCTCTGATCAAGCAGATTC 361
Query 362 CACGAATCCTCGGCCAGGTTTAAATAAGGCAGGAAAGTTCCCTTCCCTGCTCACACACA 421
Sbjct 362 CACGAATCCTCGGCCAGGTTTAAATAAGGCAGGAAAGTTCCCTTCCCTGCTCACACACA 421
Query 422 ACGAAAACATGGTGGCCAAGTGGATGAGGTGAAGTCCACAATCAAGTTCCAATGAAGA 481
Sbjct 422 ACGAAAACATGGTGGCCAAGTGGATGAGGTGAAGTCCACAATCAAGTTCCAATGAAGA 481
Query 482 AGGTGTTATGTCTGGCTGTAGCTGTTGGTCAAGTGAAGATGACAGCGATGAGCTTGTGT 541
Sbjct 482 AGGTGTTATGTCTGGCTGTAGCTGTTGGTCAAGTGAAGATGACAGCGATGAGCTTGTGT 541
Query 542 ATAAACATTCACCTGGCTGTCAACTTCTTGGTGTGCTTGTCTCAAGAAAACTGGCAGAATG 601
Sbjct 542 ATAAACATTCACCTGGCTGTCAACTTCTTGGTGTGCTTGTCTCAAGAAAACTGGCAGAATG 601
Query 602 TCCGGGCCCTTATATATCAAGAGCACCATGGGCAAGCCCGCCGCTATATTTAA 654
Sbjct 602 TCCGGGCCCTTATATATCAAGAGCACCATGGGCAAGCCCGCCGCTATATTTAA 654
```

12) Finally, the speculation in the discussion that NMD blockade might be useful therapeutically is rather vague. Most rescued transcripts would still produce defective proteins. Could the potential rescue frequency be quantified?

Inhibition of NMD would result in read-through transcripts essentially converting a stop codon to a missense mutation. So, we agree with the reviewer that the protein would potentially be defective because of the missense mutation. However, this approach may change the problem from complete loss of protein expression from an allele to expression of a “less” functioning protein. The therapeutic value of this approach has been well documented previously in cancer and non-cancer contexts such as cystic fibrosis and Duchenne muscle dystrophy. In addition, re-expression of a mutated protein would be expected to stimulate an immune response against the tumour as previously shown. We have now added references to support the statement that is mentioned in the discussion:

“our work unravels unrecognized dependencies of tumours on NMD-mediated loss of function with potential therapeutic opportunities {BartonDavis:1999hg, Bedwell:1997uj, Floquet:2011hd, Kashima:2006cr, Bruno:2011gr}”.

Reviewer #2 (Remarks to the Author):

“A pan-cancer genome-wide analysis reveals tumour dependencies by induction of nonsense mediated decay”.

The paper is touching on an important subject, predicted loss of function mutation and nonsense mediated decay.

Overall. Not seeing the author name is ok. However, by judging by the number of unclear facts, the manuscript would benefit by having either more authors going over the manuscript.

1-In their treatment of the question, the author(s) remains unclear sometime talking about causation, sometimes talking about association.

We modified the statements. Most of our findings from the data should refer to associations but imply causation that need to be validated experimentally.

2-The study is based on the pre-existing TCGA dataset, but almost no explanation of what the author used is provided.

We downloaded the pan-cancer data from UCSC Xena and Broad GDAC Firehose. (The original link to UCSC Xena was <http://tcga.xenahubs.net>, but it changed to xenabrowser.net. We have updated this in the revised manuscript.) We also clarified which files were used and which version was utilized the first subtitle “Data pre-processing” of the Methods. We cited the source of data, which contains the detailed explanation of each dataset. This paragraph has been modified in Methods:

“The following files: 1) RNA-seq data (tcga_RSEM_Hugo_norm_count, 2016-02-18), 2) somatic mutations annotation file (MAF, 2016-04-28) for results of DNA sequencing, 3) CNV data (Gistic2_CopyNumber_Gistic2_all_thresholded.by_genes, 2016-08-16) and clinical data (PANCAN_clinicalMatrix, 2016-04-30) were downloaded from the “TCGA Pan-Cancer (PANCAN)” cohort at the TCGA hub and the GA4GH-BD2K (TOIL) hub on USCS Xena (<https://xenabrowser.net>). The TCGA Pan-cancer RNA-seq data contained normalised and log transferred counts, which were quantified by RSEM. We downloaded the non-negative matrix factorization (NMF) cluster data (Version 2016_01_28) from the Broad GDAC Firehose (<http://gdac.broadinstitute.org>), which clustered samples based on mRNA-seq data.”

3-The authors mention 24 cancer types but TCGA has a number of other cancer they study, the author(s) should mention how the selection has been done.

We have added the explanation in the Methods under Data pre-processing. We chose 24 out of 32 cancer types. We only chose cancers that had more than 10,000 somatic mutations in total so that sufficient samples are available to enable adequate statistical analysis. This sentence has been added in Methods:

“To obtain sufficient samples for adequate statistical analysis, we only analysed the 24 cancer types that had more than 10,000 somatic mutations in total.”

Cancer	Number of total mutations
skin cutaneous melanoma	422553
lung adenocarcinoma	244440
uterine corpus endometrioid carcinoma	184861
stomach adenocarcinoma	148520
bladder urothelial carcinoma	135077
colon adenocarcinoma	125522
head & neck squamous cell carcinoma	115338
breast invasive carcinoma	90490
lung squamous cell carcinoma	65305
liver hepatocellular carcinoma	53777
cervical & endocervical cancer	46547
esophageal carcinoma	41378
rectum adenocarcinoma	34259
sarcoma	30391
pancreatic adenocarcinoma	30355
brain lower grade glioma	27721
kidney clear cell carcinoma	26000
glioblastoma multiforme	25131
ovarian serous cystadenocarcinoma	24092
kidney papillary cell carcinoma	16951
diffuse large B-cell lymphoma	16918
adrenocortical cancer	13129
thyroid carcinoma	13084
prostate adenocarcinoma	12348
uterine carcinosarcoma	9149
mesothelioma	8577
thymoma	6326
testicular germ cell tumor	5598
cholangiocarcinoma	4228
pheochromocytoma & paraganglioma	4088
kidney chromophobe	3835
uveal melanoma	2174

4-The data used from TCGA are mainly the DNA and RNA sequencing experiment but this is almost never mentioned.

We apologize for this oversight. We have now addressed this under point number 2.

5-Also, the majority of these mutations are likely to be somatic in origin and this is not much discussed. Somatic mutations can come with different allelic ratio in the tumour sample and this has specific consequences.

- a- We have stated the somatic mutations several times in the manuscript, such as “*We applied our prediction algorithm to all reported somatic mutations from 24 cancers*” in the first paragraph of Results. In addition, we added the “somatic” to the first paragraph of Methods.
- b- We thank the reviewer for raising this point. In the work of Lehner et al., published recently in Nature Genetics “The rules and impact of nonsense-mediated mRNA decay in human cancers”, the allele frequency was found to only explain 0.4% of the effect of NMD-elicited mutations on expression levels. However, we believe that the combination of the allelic ratio, allele-specific expression and the prevalence of a mutation is important in predicting the effect of mutations on expression. However, this information is not currently available in TCGA data. More comprehensive public databases and the application of more advanced sequencing technologies can help solve the question in the future. We have added the following sentences in the discussion:

“Another unaddressed question is the relationship between the mRNA level and the combination of allelic ratio, allele-specific expression and the prevalence of a mutation. The future application of advanced sequencing technologies and the development of novel computational models would enable further detailed analyses of the effect of a mutation on gene expression.”

6- Page 1: TSG should be defined in full before the acronym is used

The full definition and acronym have been added in the abstract as follows:

“We found that NMD-elicited mutations in tumour suppressor genes (TSGs) are associated with...”

7-Page 2 Ch 2: the authors mention a validation; they should spell out if it an n experimental one

The validation is an independent dataset other than the TCGA data, which was published as “Wang, K. et al. Whole-genome sequencing and comprehensive molecular profiling identify new driver mutations in gastric cancer. *Nat Genet* 46, 573–582 (2014)”. We have added the explanation under “Methods, NMD-elicited mutations in hypermutation”:

“The validation dataset was from an independent dataset¹³ other than the TCGA data.”

8-Page 3 Line 6: The author mention “unexpectedly” but they should mention if they expected more or less.

Many studies simply treated the frameshift mutations and nonsense mutations as NMD-elicited mutations; therefore, the expectation was 100%. However, our study showed that only two-thirds of these two types of mutations can elicit NMD, which was, therefore, unexpected. The word “only” in the following sentence should clarify this issue:

“Unexpectedly, our analysis indicated that only two thirds of the mutations that were annotated.....”

9-Page 1 line 7 when mentioning that 6% of the million mutation are NMD-elicited. I am not sure this number is relevant

This gave a general overview of the proportion of NMD-elicited mutation, which is non-negligible. The percentage is not “that relevant” but we think that it is important to mention it so that readers can obtain a global view of the effect of NMD.

10-Page 3 chapter 2. When the author mentions expression level, they mention lower expression. The authors should clarify if these if they mean it in an absolute or relative manner. It is clear from the figure that it is relative but the sentence is ambiguous.

It is the relative expression. To clarify, we have added this sentence:

“Median ratio of relative expression of variant (REV, see Methods) = 0.54, $P < 2.2e-16$, one-side t test and Fig. 1d).”

11- Page 3 Chapter line 6

When tests on stomach and kidney are mentioned the authors quote P value threshold of 0.007. It is not clear if the authors are taking into account multiple testing.

We thank reviewer for the suggestion. However, in this case we have not done multiple testing. We have fitted a generalized linear model for the regression of mutation numbers on tumour types and then tested if the associated linear coefficient was the same for all cancers.

12-Page 3 last sentence of chapter 2.

It is not clear to me that the results presented are indicating a contribution of loss of function.

It could just be a “passenger” event at this stage of the evidence presented.

Thank you for raising the point. It is important to curate each mutation critically. Please note that we used the word “suggest” and not “confirm” because we are aware of the limitation. However, a substantial reduction of gene expression would reasonably be expected to be associated with loss of function. For example, our results showed an association between NMD-elicited mutations and a significant reduction in the expression levels of the two important tumour suppressor genes, *TP53* and *NFI*. Previous studies provided strong evidence on the relationship between the down-regulation of these genes and the tumorigenesis. Therefore, it

is highly unlikely that reduction in the expression of such TSGs is a passenger event. Please note the following in the results section:

“We observed that not all NMD-elicited mutations were associated with a similar magnitude of reduction in gene expression. In addition, NMD-elicited mutations seemed to target particular genes at higher frequency than others. For example, the two genes that were most widely affected by NMD-elicited mutations were the tumour suppressor genes (TSGs) TP53 (23 cancer types affected) and NF1 (22 cancer types affected).”

13-Page 3 chapter 3-the number of studied TSG genes should be quoted

In the same chapter the author mentioned: “mutations resulted in a significant reduction in gene expression”. There is no evidence that the lower expression is a consequence of the mutations. It is just an association. And should be described as such

a- Number of studies TSGs is 71. We have added the sentence in the Methods:

“The number of studied TSGs is 71.”

b- We have replaced “result in” with “are associated with”.

14- Page 4 line 5: 29%. IS it out of all n=XXX samples? In that case the number should be given.

We have added the numbers as follows:

“The overall prevalence of NMD-elicited mutations in tumour suppressor genes (TSGs) was 29% (2206/7725; range: 5%-60%).”

15- Page 4 chapter 2. The authors mention the occurrence of mutations and deletion. The author should describe how large of a deletion they could detect. Then when they perform an assessment of expression, should they look at allele specific expression or at least discuss the limitation if they cannot.

We thank the reviewer for this suggestion. In the TCGA data, copy number was measured by whole-genome microarray and estimated by GISTIC2 at the gene level, as we have mentioned in the Methods.

We believe that the allele-specific expression is potentially important. We have now added new sentences in the discussion to address this point and also the point mentioned in question number 5:

“Another unaddressed question is the relationship between the mRNA level and the combination of allelic ratio, allele-specific expression and mutation prevalence. The future application of advanced sequencing technologies and the development of novel computational models would enable further detailed analyses of the effect of a mutation on gene expression.”

16- Spell out MWW when first mentioned

The full name of the test has been added when first mentioned as

“NMD-elicited mutations in these genes are associated with a significant reduction in gene expression (median ratio of REV = 0.07 for NF1, 0.06 for TP53; $P < 2.2e-16$, one-side Mann-Whitney-Wilcoxon (MWW) test, and Fig. 2a).”

17- Table 1: the acronym of Cancer types is not easy for reader. Mention organ.

We added the full names of cancers in Table 1. This can help distinguish between the cancers in the same organ, such as kidney clear cell carcinoma and kidney papillary cell carcinoma.

18- Author mentioned, “profound reduction in gene expression at TSG is most probably the result of the mutated allele and deletion of the wild type allele”. Can they assess it by allele specific expression?

Thank you for raising this point. We compared the relationship between NMD-elicited mutations and the expression levels of TSGs in the presence or absence of a deletion. We observed that NMD-elicited mutations are associated with a significant reduction of gene expression when compared to other types of mutations independent from whether or not there is an associated deletion of an allele, see the figure below. We have now added this comparison as supplementary figure 3b and added the following text in the results section:

“Importantly, in the absence of a deletion of an allele, NMD-elicited mutations are still associated with a significant reduction of gene expression but at a lower magnitude compared to cases when a deletion is present ($P < 2.2e-16$, MWW test, fold change = 0.43 and $P < 2.2e-16$, MWW test, fold change = 0.09, respectively, Supplementary Fig. 3b).”

The point about allele-specific expression has been addressed in relation to point number 15.

19- Discussion: number and overview in early stage of discussion should also be present in the Introduction or early part of the results.

Thank you for the suggestion. The numbers and overview are in the first part of the Results on the page 3. We have added the overview of the results in the last paragraph of the introduction and removed this from discussion.

20- Figure 1: The use of panel is a bit excessive in panel B of figure 2, nothing is readable

We moved the Figure 2B to Supplementary Figure 1 to increase its size and readability.

21- Page 7 chapter 2 sentence 2. The author talks about efficiency that directly imply causation when talking about TSG, whereas it is clearly stated that the two hits hypothesis with co-occurrence of deletion is a possible explanation.

I suggest removing the term NMD efficiency

Thank you for the suggestion. We have changed it to “the magnitude of associated reduction in expression (MARE)”.

22- In table 1: we note that a lot of gene such as Titin and Mucin, correspond to some of the biggest and most polymorphic gene in the genome. Would correcting by gene size necessary?

We agree with the reviewer and we have, therefore, marked those genes in the table as potential false positive cancer associated genes. However, we feel that adjusting for gene size may introduce significant biases. We have, therefore, opted not to do this normalization.

Reviewer #3 (Remarks to the Author):

This manuscript assesses the prevalence and impact of NMD-provoking mutations in the TCGA data set. These authors use a previously derived algorithm to identify ~75,000 NMD provoking mutations, and assess how affect RNA expression from the TCGA data, and their enrichment in tumor suppressor genes, frequently mutated genes in cancer, and amongst distinct cancers.

This is an interesting paper and will be of interest to those studying NMD and those in the cancer field. However, it suffers from some deficiencies inherent in their methodology.

1. Their rule for identifying NMD provoking mutations is likely not complete nor as sensitive or specific as they would like. Although they did find decreased mRNA expression in those transcripts that fit their rule in general, it is likely that other mutations, deletions/insertions in

the 3'UTR also lead to NMD, and that some of these mutations identified as NMD provoking are not degraded by NMD.

This is a valid point. We have now added this in the discussion.

“Although the rules that we used to predict NMD-elicited mutation are known to have a significant effect on mRNA levels, other potentially important factors may have been overlooked because of the incomplete understanding of the mechanisms of NMD in humans. For example, the potential role of insertions and deletions in the 3'UTR in inducing NMD has remained unclear.”

2. It is unclear the contribution of mutations in members of the NMD complex (described by the Wilkinson and recently the Steitz groups) and/or suppression of NMD by the tumor microenvironment (described by the Gardner group) may impact their findings.

- a- We attempted to study the effect of the interaction between mutations in members of the NMD complex and NMD-elicited mutations in genes that are frequently affected by them such as *TP53*. Unfortunately, this was not possible because the low number of mutations affecting the NMD complex precluded any meaningful global analysis. This was also confounded by the inability to conclude whether or not a mutation in the NMD complex would be associated with loss of function. We have added the following in the discussion section:

“Whether or not mutations in members of the NMD complex may have an impact on eliciting NMD is difficult to study on a global scale because of the low number of mutations in this complex and the lack of clarity as to whether any particular mutation in this complex is associated with loss of function.”

- b- The suppression of NMD by the tumor microenvironment is very interesting. The following sentences have been added to discussion and the related papers have been cited.

“In addition, our results do not take into account the potential suppression of NMD by the tumour microenvironment as recently reported^{34,35}”

3. The role of mutations in the promoter, transcription factors, epigenetic factors, and/or non-coding mRNAs likely has a great impact on mRNA expression, and is difficult to take into consideration.

Yes, it is. Other factors that can affect mRNA expression are important. However, we have used appropriate statistical methods to show that NMD-elicited are associated with a significant reduction in gene expression. We have added the sentences to discussion to discuss our limitation and perspective:

“Other than NMD-elicited mutations, mutations occurring in transcription factors, epigenetic factors and non-coding mRNAs may affect mRNA levels.”

4. While interesting, it is difficult to know the significance of differences found in tumor types, and perhaps this could be better addressed in the discussion.

We have now added the following in the discussion:

“Whether this higher load of NMD-elicited mutations is the result of specific mutational signatures that occur in these particular tumours is testable in future studies. In addition, further analysis of the impact of cancer-specific mutational patterns²⁷ and signatures on potential enrichment of NMD-elicited mutations in particular pathways might provide important insights into the possible dependencies of these tumours on NMD.”

Reviewers' Comments:

Reviewer #1:

Remarks to the Author:

The manuscript is significantly improved and generally has responded adequately to the critiques. The data will be of interest to the cancer genomics community. One comment remains.

Regarding the possibility of "read-through" therapy, the very brief comments in the results and discussion are still bothersome. It is nice to hope that this may be possible someday as proven for specific mutations in a few germ line genetic diseases. However, the comment lacks nuance, since, while true that some read through transcripts might be fully functional and others merely hypomorphic, it is also possible that they may become dominant negative or result in a deleterious impact on the balance of alternative transcripts. Each mutation would have to be individually evaluated for these risks since each "read-through" product would have its own properties. For this reason, I think the existing comments should either be deleted or adjusted to account for these and other complexities that can and will arise in attempts at therapeutic development.

Reviewer #2:

Remarks to the Author:

The authors have now responded to the points that I raised. Overall the manuscript is clearly improved and less ambiguous.

Response to reviewers' comments

Reviewer #1 (Remarks to the Author):

“The manuscript is significantly improved and generally has responded adequately to the critiques. The data will be of interest to the cancer genomics community. One comment remains.”

“Regarding the possibility of "read-through" therapy, the very brief comments in the results and discussion are still bothersome. It is nice to hope that this may be possible someday as proven for specific mutations in a few germ line genetic diseases. However, the comment lacks nuance, since, while true that some read through transcripts might be fully functional and others merely hypomorphic, it is also possible that they may become dominant negative or result in a deleterious impact on the balance of alternative transcripts. Each mutation would have to be individually evaluated for these risks since each "read-through" product would have its own properties. For this reason, I think the existing comments should either be deleted or adjusted to account for these and other complexities that can and will arise in attempts at therapeutic development.”

Thank you for raising this very important point. We have deleted or adjusted existing comments that oversimplified the complicated context of the “read-through” therapy. We also added a reference to support the statement of potential therapeutic advantages via expression of new epitopes that can stimulate antitumour immune response as previously published in *Nature* (DOI: 10.1038/nature08999).

In the discussion, we made the following modifications:

“We also found evidence of tumour type-specific targeting of driver TSGs by NMD (Table 1) suggesting that therapeutic targeting of NMD could be successful in a wide range of tumours. In addition, inhibition of NMD can result in the expression of new antigenic epitopes and enhanced anti-tumour immune response. However, other complexities need to be considered. For example, the read-through product that is generated from a PTC-harboured transcript may result in a negative dominant effect or other deleterious impacts.”

Reviewer #2 (Remarks to the Author):

The authors have now responded to the points that I raised. Overall the manuscript is clearly improved and less ambiguous.

We thank the reviewer for the previous suggestions.